CD4 count and tuberculosis risk in HIV-positive adults not on ART: a systematic review and meta-analysis

Ellis Penelope K. 1
Martin Willam J. 1
Dodd Peter J. p.j.dodd@sheffield.ac.uk 2
1 Sheffield Medical School, University of Sheffield , United Kingdom
2 School of Health and Related Research, University of Sheffield , United Kingdom
Flores-Valdez Mario Alberto
Electronic publication date: 2017 Dec 14
Publication date: 2017
Volume: 5
Electronic Location ID: e4165
Received 2017 Sep 25; Accepted 2017 Nov 23
Copyright: ©2017 Ellis et al.
Copyright year: 2017
Copyright holder: Ellis et al.
License: This is an open access article distributed under the terms of the Creative Commons Attribution License, which permits unrestricted use, distribution, reproduction and adaptation in any medium and for any purpose provided that it is properly attributed. For attribution, the original author(s), title, publication source (PeerJ) and either DOI or URL of the article must be cited.
License URL: https://creativecommons.org/licenses/by/4.0/

Keywords: TB, Antiretroviral therapy, Modeling, Opportunistic infections, CD4 cell count

Funding: UK MRC MR/P022081/1 This work was partially supported by the UK MRC (MR/P022081/1). There was no additional external funding received for this study. The funders had no role in study design, data collection and analysis, decision to publish, or preparation of the manuscript.

==============================
Background

CD4 cell count in adults with human immunodeficiency virus (HIV) infection (PLHIV) not receiving antiretroviral therapy (ART) influences tuberculosis (TB) risk. Despite widespread use in models informing resource allocation, this relationship has not been systematically reviewed.

Methods

We systematically searched MEDLINE, Aidsinfo, Cochrane review database and Google Scholar for reports in English describing TB incidence stratified by updated CD4 cell count in cohorts of HIV-positive adults (age ≥15 years) not on ART (PROSPERO protocol no: CRD42016048888). Among inclusion criteria were: reporting precision for TB incidence, repeated CD4 measurements, and TB incidence reported for those not on ART or monotherapy. Study quality was assessed via the Newcastle-Ottawa tool for cohort studies. A Bayesian hierarchical model was fitted to estimate the pooled factor increase in TB incidence with respect to CD4 cell count decrement.

Results

A total of 1,555 distinct records were identified from which 164 full text articles were obtained. Common reasons for exclusion of full texts were: no valid TB incidence, no repeat CD4 measurements, and not reporting TB incidence by ART status. The seven studies included reported on 1,206 TB cases among 41,271 individuals, with a typical median follow-up of four years. Studies were generally rated as moderate or high quality. Our meta-analysis estimated a 1.43 (95% credible interval: 1.16–1.88)-fold increase in TB incidence per 100 cells per mm3 decrease in CD4 cell count.

Discussion

Our analysis confirms previous estimates of exponential increase in TB incidence with declining CD4 cell count in adults, emphasizing the importance of early ART initiation to reduce TB risk in PLHIV.

Introduction

Tuberculosis (TB) and the human immunodeficiency virus (HIV) are the two leading infectious killers with an estimated 1.4 million deaths from TB (excluding those with HIV) and 0.4 million deaths in those with HIV and TB in 2015 (World Health Organization, 2016). HIV is the most potent known risk factor for TB disease incidence and has been a major factor in the upswing in TB incidence in sub-Saharan Africa throughout the 1990s. Around 11% of the 10.4 million incident TB cases globally and around 20% of total global mortality from TB were estimated to be in people living with HIV (PLHIV) in 2015 (World Health Organization, 2016). TB is implicated in 36% of deaths from HIV (UNAIDS, 2016).

Anti-retroviral therapy (ART) for in PLHIV is known to be around 70% protective against TB incidence (Suthar et al., 2012). Isoniazid preventive therapy (IPT) against TB is now recommended for 36 months in PLHIV (World Health Organization, 2015b), and further reduces the risk of TB by 30–50% (Golub et al., 2007; Rangaka et al., 2014; Samandari et al., 2011; Yirdaw et al., 2014). ART additionally has substantial benefits of reduced all-cause mortality (Antiretroviral Therapy Cohort Collaboration, 2008). Despite encouraging recent progress however, under 50% of adults living with HIV are on ART (UNAIDS, 2016), and under 40% of PLHIV newly enrolled in care received IPT in 2015 (World Health Organization, 2016).

CD4+ cells play a pivotal role in both HIV and Mycobacterium tuberculosis (M.tb) infection, and the peripheral blood CD4+ lymphocyte count (CD4 cell count) played an important role historically in ART initiation guidelines and has been shown to be a strong predictor of TB risk. Despite the fact that HIV treatment guidelines recommend ART initiation of regardless of CD4 count (World Health Organization, 2015a), individuals have typically undergone some CD4 decline by the time they are diagnosed and begin HIV treatment. Understanding the population-level implications of this increase in mean risk of TB is therefore still crucial to understanding the epidemiology of a given setting, predicting the impact of interventions to improve HIV diagnosis, and resource planning. Previous analyses (Williams & Dye, 2003; Williams et al., 2010) have found an exponential relationship between CD4 decrement in PLHIV not on ART and TB disease incidence, and their analysis has been widely used since in modeling analyses of the impact of HIV interventions on TB, linked HIV/TB resource planning (e.g., the OneHealth Tool frequently used to support applications to the Global Fund), and is used in TB burden estimation (Pretorius et al., 2014a; Pretorius et al., 2014b; Williams & Dye, 2003; Williams et al., 2010). However, this relationship was not based on a transparent systematic review of available data.

Given the importance of the quantitative correspondence between CD4 count and TB risk in PLHIV not on ART and the absence of a systematic evidence synthesis, we sought to provide a meta-analytic estimate of this relation for adults based on systematically identified cohorts of ART-naïve PLHIV reporting TB incidence by updated CD4 count.

Methods

This systematic review is reported in accordance with the Preferred Reporting Items for Systematic Review and Meta-Analyses (PRISMA) (Moher et al., 2009). The protocol was registered with the International Prospective Register of Systematic Reviews (identification number: CRD42016048888).

Cohort definitions, selection criteria, and search strategy

We sought cohorts of HIV-positive adults (≥15 years of age) who were not on ART, with HIV infections and data stratified by CD4 count (exposure) at TB incidence (outcome). Studies were also included if they reported relative measures of TB incidence across CD4 categories, such as hazard ratios. The analysis comparison was between TB incidences for different CD4 count categories.

To be eligible, a study had to present empirical data on at least five cases of TB in adults. Studies were excluded if they explicitly restricted to groups not generally representative of adult TB cases in that population at that time, e.g., studies that only addressed one clinical manifestation of TB; that only comprised of hospitalized TB cases; studies that focused on migrant populations. Where explicit use of isoniazid preventive therapy was stated, those receiving IPT were excluded (similarly for any other form of prophylaxis). To allow appropriate study weighting, TB incidence or relative TB incidence had to be reported with confidence intervals, or as person-time and the number of incident TB cases.

To be eligible, studies had to report separately on individuals not on ART (ART including monotherapy etc.). Studies where the majority of HIV infections are known to be not HIV-1 were excluded. (HIV-1 is the most common major type of HIV.) We required at least one CD4 count measurement subsequent to the baseline measurement at enrolment.

If the same cohort was published more than once, or individuals were described in more than one cohort, studies were only included once.

We searched MEDLINE (11/16) for articles published in English at any time. (See Appendix S1 for full search strategy.) The search was developed in consultation with an information specialist and sensitivity assessed by checking against known papers from Williams’ analyses (Williams & Dye, 2003; Williams et al., 2010). We also searched the Aidsinfo database for registered studies that may have had relevant publications listed, and the Cochrane database of systematic reviews for relevant reviews.

The references of included studies were also searched for additional studies, as were articles listed as citing these articles on Google Scholar.

Data extraction and risk of bias

Two reviewers (PKE & WJM) independently screened articles returned by the search, and discrepancies were resolved in discussion with a third reviewer (PJD). Reasons for rejection were recorded.

A data extraction form was developed based on previous experience with a related review (Dodd et al., 2017), and refined after application to the first included article. Data was independently extracted (by PKE & WJM) on: first author, publication year, study country, study years, study description, ethnicity, CD4 cell count category, number in cohort, number of TB cases, crude and adjusted hazard ratios, patient years, TB incidence (and 95% confidence interval [CI]), frequency of CD4 cell count measurement, proportion of TB bacteriologically confirmed, Bacille Calmette Guérin (BCG) vaccination coverage, tuberculin skin test (TST) status, age range, sex and HIV epidemic type. Any discrepancies were resolved by discussion with a third reviewed (PJD).

The 95% Poisson exact confidence intervals based on reported event counts and person time were used to define precision for incidence data points. Where only adjusted hazard ratios with respect to a given CD4 category were reported, the stated 95% CIs were used to estimate precision. For the single TB incidence data point with zero TB events, we used the lower bound of the Poisson exact confidence interval divided by 100 as the central incidence estimate (as the logarithm of zero is not defined), and examined sensitivity to this approximation by repeating our analysis with an incidence point estimate value 10 times smaller again. CD4 category mid-points were used in statistical analyses with the assumption of an upper CD4 count of 1,000 cells/mm3 where not stated.

To assess risk of bias in individual studies, the Newcastle-Ottawa instrument for cohort studies (Wells et al., 2000) was adapted (see Appendix S1) and summarized as low-, moderate-, or high-quality for our question on selection, comparability and outcome domains depending on whether few, some, or most of the question items in a domain were answered positively.

Statistical analysis

In order to use information on both TB incidence and hazard ratios, we performed a Bayesian hierarchical meta-analysis assuming that the logarithmic TB incidence for a given CD4 category was proportional to a study-specific intercept (capturing the background TB incidence rate) and a study-specific gradient with respect to CD4 cell count drawn from a common distribution (analogously to a frequentist random-effects model). The model was fitted with Markov chain Monte Carlo (MCMC) methods in the R environment for statistical computing (R Core Team, 2016) using RStan (Stan Development Team, 2016) (see details and code in Appendix S1). Gelman–Rubin convergence statistics computed and chains plotted.

Heterogeneity was calculated as τ2, the variance of the summary distribution for the log-gradient, and a corresponding I2 statistic evaluated using the summary statistic of Higgins & Thompson (2002) for individual studies’ posterior variance. Forest plots of the individual and summary gradients with respect to CD4 were produced, and a funnel plot to assess risk of publication bias.

We computed the implied factor increase in TB incidence per 100 cell per mm3 by our estimate, and similarly the incidence rate ratio for TB implied at different CD4 counts.

Results

See Supplemental Information 3 for location of items reported.

Study selection and characteristics

Database searching returned 1,824 records, which together with 5 records identified by other means, led after screening to the full texts of 150 articles being examined (see flow diagram, Fig. 1). The most common reason for exclusion at the full text stage was TB incidence not being reported (with an implied measure of precision); use of ART (or monotherapy) in the cohort with TB incidence measures without stratified reporting by ART status; and only CD4 at baseline being measured. The studies used in previous analyses (Williams & Dye, 2003; Williams et al., 2010) were captured by this strategy, but rejected for only measuring CD4 at baseline or not reporting TB incidence in PLHIV not on ART. Data from seven studies was finally extracted for meta-analysis.

Figure 1 PRISMA flow diagram of review process.

The seven studies included are summarized in Table 1 (Assebe et al., 2015; Collins et al., 2015; Grant et al., 2009; Markowitz et al., 1997; Monge et al., 2014; Nicholas et al., 2011; Wolday et al., 2003). Studies included reported on 1,206 TB cases among 41,271 individuals, with a median over median follow-up times of 4.0 years. Three studies were based in sub-Saharan Africa (with heterosexual transmission predominating), one in Haiti, and the rest in the US or Europe (mostly men-who-have-sex with men, or injecting drug use), with most of the data being from the late 1990s or early 2000s. The more recent studies included evaluations of ART timing and IPT. The largest study was a retrospective observational cohort in the UK (Grant et al., 2009). Less than half the studies reported (three of seven) TST status, with the proportion of patients in cohorts being TST positive ranging between 6% and 38%. The proportion of TB bacteriologically confirmed (reported by four of seven studies) ranged between 50% and 71%. Only one study (Wolday et al., 2003) reported BCG vaccination status (38% of PLHIV).

Table 1 Summary of studies.

First author, year	Country	Years of study	Study description	Study conclusion	Number in cohortg	Ethnicity	CD4 count category	Number of TB cases	Patient years	TB incidence per 1,000 person-years	Bacteriologicaly confirmed TB (%)	TST positive (%)	Age range (median)	Male (%)	Epidemic type	Qualitya (selection/ comparability/outcome)	
Assebe, 2015	Ethiopia	2008–2012	Retrospective cohort study of patients in pre-ART care at a tertiary hospital focusing on the effect of IPT. Cohort comprised IPT (n = 294) and non-IPT (n = 294) patients sampled from 3,476 in pre-ART care. Median follow-up 24.1 months. Adjusted hazard ratio from Cox PH model reported for CD4 categories (adjusting for IPT use) based on n = 585 patients.	IPT found to halve TB incidence.	588	Black African	<350	49	1297.5	aHR= 3.16 (1.04–3.92)	–	–	15–64	38	Heterosexual	B/A/B	
350–499	aHR = 2.87 (1.37–6.03)	
≥500	aHR = 1g	
Collins, 2015	Haiti	2005–2012	Open-label RCT, using the CIPRA-HT001 cohort to receive early ART (n = 408) or delayed ART (i.e., when CD4 < 200; n = 408). Median follow-up of 4.4 in the early treatment arm and a median 3.9 years in the delayed treatment arm. 43 participants with prevalent TB at initiation excluded from incidence analysis, which reported an adjusted HR of TB incidence per 50 cells/mm3 CD4 decrement from Cox PH models (this included person-time on ART). Model including adjustments for treatment group and PPD-status/IPT used.	Delayed ART initiation results in persistent increased TB risk.	816	–	–	96	aHR = 1.24 (1.11–1.38) per 50 CD4 cells/mm3 decline	61	27e	≥18 (40)	42	–	B/A/A	
Grant, 2009	UK	1996–2005	An observational cohort study of patients (UK Collaborative HIV Cohort, restricted to those who joined prior to 2006); median follow-up 4.2 years. TB within 3 months of enrolment excluded, as were individuals without CD4 count or who were lost to follow-up or died within 3 months of first visit. TB incidence reported separated for those not taking ART.	Early HIV diagnosis and IPT key to reducing the particularly high TB incidence in non-white & low-CD4 PLHIV in the UK.	22,833	Black African	<50	9	176	51.1	–	–	≥16 (34)	36	Heterosexual	A/B/B	
50–199	13	612	21.2	
200–349	14	1,453	9.6	
350–500	7	1,443	4.9	
>500	7	1,553	4.5	
White	<50	9	850	10.6	≥16 (35)	92	MSM	
50–199	25	2,631	9.5	
200–349	20	6,636	3	
350–500	9	7,548	1.2	
>500	3	9,560	0.3	
Other	<50	3	252	11.9	≥16 (34)	77	MSM	
50–199	4	535	7.5	
200–349	5	1,422	3.5	
350–500	11	1,587	6.9	
>500	1	2,180	0.5	
Markowitz, 1997	USA	1988–1990	Prospective, multi-center cohort study of HIV seropositive patients (the Pulmonary Complications of HIV Infection Study) representative of US PLHIV; median follow-up of 53 months. Around 4% of patients were taking IPT. TB incidence increased with PPD induration.	TB incidence highest in patients with CD4 <200 and PPD patients.	1,130	White/Black /Hispanic	<200	17	1,417	12	71	6d	18–67 (37b)	87	MSM/IDU /heterosexual	C/B/A	
≥200	14	2,800	5	
Monge, 2014	Spain	2004–2010	Open, multi-center, prospective cohort of ART naïve patients (CoRIS = Spanish AIDS Research Network Cohort). Around half of TB cases were in people on ART. Adjusted rate ratios by CD4 category from Poisson regression (adjusting for ART-status) were therefore used. TB episodes during follow-up and up to 3 months prior to enrolment were included.	Early HIV diagnosis and ART should be used to help address high TB incidence in PLHIV.	6,811	–	<200	124	aHR = 5.20(3.25–8.33)	65	–	>13	80	Heterosexual /IDU	B/A/C	
200–350	32	aHR = 1.54(1.01–2.34)	
>350	37	aHR = 1g	
Nicholas, 2011	Guinea, Kenya, Malawi, Mozambique, Nigeria & Uganda	2006–2008	Multi-center, retrospective cohort study based on the FUCHIA database from all sub-Saharan Médicins sans Frontières HIV programmes. TB at, or within 15 days of, enrolment excluded; patients with <15 days of follow-up excluded. Median follow-up 9 months. TB recorded pre-ART and during ART. Median pre-ART follow-up of 9 months.	Importance of early HIV diagnosis and treatment and implementation of the 3Is.	8,998	Black African	<50	59	98	602	–	–	27–40c (33)	27	Heterosexual	A/A/B	
50–99	68	170	401	
100–199	162	732	221	
≥200	398	6,180	64	
Wolday, 2003	Ethiopia	1997–2001	A prospective cohort study (Ethio-Netherlands AIDS Research Project = ENARP) to study biomarkers associated with TB progression. HIV-positive and negative factory workers with an overall median follow-up of 3.8 years and 6-monthly assessments.	Low CD4 and high viral load associated with TB incidence; successful TB treatment does not reduce viral load.	95	Black African	<200	5	46	107.6	50	38d	(34)	52	Heterosexual	C/A/B	
200–499	5	121	41.2	
≥500	0	43	0	
Notes.

a Quality assessed with a modified Newcastle-Ottawa scale for cohorts with A/B/C according to all/some/few criteria met in each domain.

b Mean.

c Inter-quartile range.

d 5 mm cut-off used in tuberculin skin test.

e Cut-off used in tuberculin skin test not stated.

f Reference value.

g Cohort number restricted to those included in TB incidence calculation.

aHR adjusted hazard ratio

ART antiretroviral therapy

IPT isoniazid preventive therapy

TB tuberculosis

RCT randomized controlled trial

PPD purified protein derivative

TST tuberculin skin test

PLHIV people living with HIV

IDU injecting drug users

MSM men who have sex with men

Most studies (five of seven) reported person-time and incident TB cases by CD4 count; the two that did not (Assebe et al., 2015; Collins et al., 2015) both reported adjusted hazard ratios for CD4 categories derived from Cox proportional hazards models. In addition, around half of the incident TB cases reported in one study (Monge et al., 2014) were on ART, and so we used the adjusted hazard ratio for this study (which adjusted for ART status) in the meta-analysis. The studies analyzed as reporting TB incidence (four of seven) contributed 868 incident TB cases during 50,045 person-years of observation. While the background TB incidence rates varied substantially between settings, the trends in TB incidence by CD4 count within study qualitatively supported the assumption of a linear relation to logarithmic TB incidence (see Fig. 2). CD4 categories were not the same across studies, and typical intervals between measurements also varied.

Figure 2 Incidence of tuberculosis in adults living with HIV (age ≥15 years) not on antiretroviral therapy, by study and CD4-positive lymphocyte count.

Study data reported as incidence are plotted as triangles with confidence intervals; study data reported as hazard ratios are reported as dots, with confidence intervals except for the reference category. Note: the rightmost data point for Wolday, 2003 had zero TB cases which is not defined on a logarithmic scale—only the top of the Poisson exact confidence interval is shown, with the bottom truncated.

The studies were rated as having high comparability, since the comparison was between subgroups of the same cohort (see Table 1). Quality around outcome measurements was also generally good; likewise for selection of individuals (representativeness of PLHIV in given country, ascertainment of HIV status). The funnel plot (see Appendix S1) showed 3 of the seven studies fell just outside the lower funnel boundary; formal tests of publication bias were not used since the number of studies included was fewer than 10.

Statistical analysis

The Gelman–Rubin statistics for the MCMC chains (chains plotted in Appendix S1) were all ≤1.002. The estimates of individual study log-TB incidence gradients with respect to CD4 are shown in the forest plot (see Fig. 3) together with the summary estimate that was 0.36 (95% credible interval (CrI [0.15–0.63]) per 100 cells/mm3 (see Appendix S1 for table). The heterogeneity measured by τ2 was 0.048 (95% CrI [0.003–0.44]) per (100 cells/mm3)2. This corresponds to an I2 statistic of 96%. The sensitivity analysis using a point-estimate of incidence a factor of 10 times smaller for the data point with no TB events yielded a log-TB incidence gradient with respect to CD4 of 0.40 (95% CrI [0.09–0.78]).

Figure 3 Forest plot of the rate of increase logarithmic tuberculosis incidence with CD4-positive lymphocyte count in adults living with HIV (age ≥15 years) not on antiretroviral therapy.

Summary and individual study measures are based on the posteriors from the hierarchical meta-analysis.

This log-TB incidence gradient is equivalent to a 1.43 (95% CrI [1.16 –1.88])-fold increase in TB incidence risk per 100 cells per mm3 decline in CD4 count. The IRR implied by this result as CD4 cell-count declines rises from just under 3 for those with CD4 cell count over 500 cells per mm3 to over 25 for those with CD4 cell counts under 200 cells per mm3 (Fig. 4).

Figure 4 Increase in relative risk of tuberculosis incidence in adults living with HIV (age ≥15 years) not on antiretroviral therapy by CD4-positive lymphocyte count.

Thick dashed lines represent 95% credible intervals around the point estimate (thick solid line); horizontal lines represent means over depicted CD4 categories; dotted lines represent 95% credible intervals for these category means; the horizontal dashed line represents an incidence rate ratio of 1 (no change). It is assumed that individuals have a CD4 count of 1,000 cells/mm3 at the point of HIV infection.

Discussion

Our systematic review and meta-analysis confirms decline in CD4 cell count among HIV-positive adults (aged ≥15 years) not receiving ART as a strong risk factor for incident TB, with the IRR for TB increasing exponentially as CD4 cell count declines. Our meta-analysis of the gradient of logarithmic TB incidence with respect to CD4 count is very close to the estimate produced previous analyses (Williams & Dye, 2003; Williams et al., 2010) but based on a non-overlapping set of papers.

The papers included in previous analyses (Williams & Dye, 2003; Williams et al., 2010) were captured by our review but rejected on the grounds they did not separately report TB incidence off ART or only measured CD4 cell count at baseline (one of our a priori exclusion criteria). It is reassuring that our analysis reaches a very similar number from a different body of evidence, while including more patients from sub-Saharan Africa. However, in retrospect, since some of the studies we included had only infrequent CD4 measurement, e.g., Collins et al. (2015) or did not report the regularity of measurement; it is not clear that their CD4 count categories are necessarily a more robust indicator of current CD4 count than a baseline measurement in study of short duration, e.g., Antonucci et al. (1995) included in Williams & Dye (2003). Another limitation of our analysis is associated with CD4 cell count categories: these were broad and differed between studies, and for our meta-analyses we used category mid-point.

We did not consider children (aged under 15 years) in our analysis, who have very different natural histories for both TB and HIV. The relationship between TB and HIV in children is the subject of a separate systematic review and meta-analysis (Dodd et al., 2017). It is possible that a small number of individuals may have contributed person-time at age 14 to the cohort in Monge et al. (2014).

A strength of this work is that the data included was based on the results of a systematic review with a clearly defined a priori search strategy, which captured all relevant articles of which we were aware. Our search did have limitations however: it restricted to articles in English; and while it included MEDLINE, it did not include another large general database such as Embase. The quality of studies included was generally rated as high for our analysis, although our assessment tool did not capture shortcomings such as the timing of CD4 measurements. There were too few studies included to formally assess evidence of publication bias.

The prevalence of Mycobacterium tuberculosis (M.tb) infection as measured by tuberculin skin test (TST) was available for a minority of studies. This is relevant because infection with M.tb is thought to confer protection against incident disease from re-infection in HIV-uninfected individuals (Andrews et al., 2012), and it may be that HIV increases susceptibility to M.tb infection. The protection conferred by previous infection is often assumed to be absent in PLHIV, but quantitative evidence is lacking due to the particular problems of TST as a test for infection in this population (Ayubi et al., 2016). Ultimately, this means the relationship analyzed is between CD4 count and TB incidence either from primary progression, re-activation or re-infection. HIV and CD4 decline may impact differently on the different routes to TB disease.

The statistical heterogeneity associated with our included studies was high. However with few studies and only study-level covariates usually available, we were not able to investigate potential causes of heterogeneity. These may have included differences between populations, strains of TB or HIV, by epidemic type or transmission route, and confounding by age or sex. BCG vaccination status was not reported, except for one study (Wolday et al., 2003), and may itself have confounded results, either through variation in coverage or variation in efficacy.

Despite the shortcomings discussed above, confidence in our results in increased by the closeness to previous similar analyses, as well as with expectations from other studies not based on TB incidence in cohorts. The average predicted IRR over all CD4 counts (∼8) is not dissimilar to the early population estimates (Corbett et al., 2003) based on the prevalence of HIV in TB cases. The increase in TB risk immediately following sero-conversion has been estimated as ∼2 (Sonnenberg et al., 2005), which is compatible with the average of ∼2 over the CD4 category 500–1,000 cells per mm3.

It is noteworthy that many of the limitations we discuss above could be circumvented by an individual patient meta-analysis. This would obviate the CD4 category mid-point assumption and time since the last CD4 measurement could be used (potentially in conjunction with imputation via models of CD4 dynamics). The effects of potential confounders such as age, sex and HIV infection route could be considered. Other questions, such as the impact of incident TB disease on CD4 dynamics or the relationship between updated CD4 cell count, time on ART, and TB incidence in PLHIV who are on ART might also be accessible. This last question, in addition to its intrinsic interest, would also allow individual patient models to include more detail and heterogeneity in their representations of TB risk.

Conclusions

The high risk of TB conferred by untreated HIV infection, the large fraction of HIV deaths associated with TB, and the still substantial numbers of PLHIV who are not on treatment underscores the importance of HIV testing, prompt ART initiation and TB-preventive therapy. Our analysis affirms the quantitative relationship underlying many TB models used in epidemic projection and resource planning. Our review and analysis both highlight the potential for an individual patient meta-analysis to improve our quantitative understanding of the relationship between HIV infection and TB risk.

Supplemental Information

Supplemental Information 1 CD4 dataset for meta-analysis

Also included in R-code form in Appendix with analysis code.

Click here for additional data file.

Supplemental Information 2 Rationale for and contribution of meta-analysis

Click here for additional data file.

Appendix S1 Further detail on statistical methods, quality assessment, data, code and results

Click here for additional data file.

Supplemental Information 3 PRISMA checklist

Click here for additional data file.

Additional Information and Declarations

Competing Interests

Author Contributions

Data Availability

The authors declare there are no competing interests.

Penelope K. Ellis and Willam J. Martin performed the experiments, reviewed drafts of the paper.

Peter J. Dodd conceived and designed the experiments, performed the experiments, analyzed the data, wrote the paper, prepared figures and/or tables, reviewed drafts of the paper.

The following information was supplied regarding data availability:

The code and raw data are provided in Appendix S1.

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
