# Peer review of "CD4 count and tuberculosis risk in HIV-positive adults not on ART: a systematic review and meta-analysis"

_PeerJ, doi:10.7717/peerj.4165_

## Round 0.1 · original submission · Minor Revisions

You will find a number of suggestions that will help better describe the scope of this manuscript, so, I hope you can submit your revised version in a timely manner.

Reviewer 1 ·

Basic reporting

The paper is well written, but can be improved with minor copy-editing.

Experimental design

The study objective and rationale for this study and study design methods are well stated.

Validity of the findings

The analysis is sound, but the findings have to specifically stated that they are applicable only to HIV positive adults (>= 15 years of age) not on ART, rather than generally to all people living with HIV not on ART, since this can be misunderstood to mean both children and adults.

Additional comments

The authors have conducted a systematic review and meta-analysis to quantify the risk between CD4+ T cell count and tuberculosis among HIV-positive adults (>= 15 years of age) who are not on antiretroviral therapy. This is a valuable study that quantifies this risk in a systematic manner, and adds the the current evidence illustrating the exponential increase in TB incidence with declining CD4+ T cell count.

Here are my specific comments:

1. Lines 1-2 -- Title: Change the “people” in title to “adults”, since this study is specific to HIV-positive adults (>= 15 years of age). You can change the title to “CD4 count and tuberculosis risk in HIV-positive adults not on ART: A systematic review and meta-analysis”

2. Line 4 -- Short title: Change short title to “CD4 count and TB risk in HIV-positive adults not on ART”

3. Lines 30-32: Add “among HIV-positive adults (>= 15 years of age) not on ART” to the end of the sentence, (i.e), change sentence to “Our meta-analysis estimated a 1.43 (95% credible interval: 1.16 – 1.88)-fold increase in TB incidence per 100 cells per mm3 decrease in CD4 cell count among HIV-positive adults (>= 15 years of age) not on ART”

4. Line 33-35: Add “among HIV-positive adults (>= 15 years of age) not on ART”, (i.e), change sentence to “Our analysis confirms previous estimates of exponential increase in TB incidence with declining CD4 cell count among HIV-positive adults (>= 15 years of age) not on ART, emphasizing the importance of early ART initiation to reduce TB risk.

5. Similar to comments 1-4, please change any reference throughout the paper from people to make it specific to adults. Thereby, the analysis and findings in this paper will not be misunderstood to be applicable to all people, including children and adults.

6. Line 43-45: Be clear there are a total of 1.8 million deaths from TB, of which 1.4 million deaths are among HIV- people and 0.4 million deaths among HIV+ people.

7. Line 154-156 and Figure 4: Why is the upper CD4 T cell count of 1000 cells/mm3 used? Can’t the curve be projected/extrapolated using the available data?

8. Line 293-294: Add “except for one study (Wolday et al. 2003)”, (i.e), modify the sentence to “BCG vaccination status was not reported except for one study (Wolday et al. 2003), and may itself have confounded results, either through variation in coverage or variation in efficacy.”

9. Figure 1: Update the box “Studies included in qualitative synthesis (n=7)” to “Studies included in qualitative and quantitative synthesis (n=7)”.

10. Figure 2 legend: Change “people” to “adults (>= 15 years of age)”.

11. Figure 3: Try the plot with X axis in linear scale instead of log scale for “Increase in TB incidence per 100 CD4 cells/mm3 decrease” -- This is likely to represent this data better for understanding. If the current plot (log scale) is a better visualization of this result, than leave it as is; if not replace it with the linear scale plot.

12. Table 1: Split the “Study description” column to “Study description” and “Conclusion”. That is, add a separate column for “Conclusion”, and move this corresponding text from the “Study description” column.

13. Table 1: Monge, 2014 study -- age range is > 13 years. Is the overall systematic review representative of >=15 years of age or >=14 years of age?

14. Table 1: Nicholas, 2011 study -- Name all the 6 countries in the “Country” column.

Reviewer 2 ·

Basic reporting

in line 109 they say they excluded studies where the majority of HIV infections are known to be not HIV-1; I think it should be clarified to the reader what HIV-1 means.
In line 194 it says "The studies used previous analyses" while I think it should say "The studies used in previous analyses"

Experimental design

I think the purpose of the study is clearly stated. Methods are explained in such a way that allows other researchers to reproduce or further explore the topic.

Validity of the findings

Theay clearly state how their findigs relate to previos results in literature, explaining where their findigs confirm what was reported before, and highlighting new possible paths for further understanding the topic.

---

## Round 0.2 · accepted · Accept

I appreciate your time in revising your manuscript, and hope to see more works from you submitted to PeerJ.